# An interpretable LightGBM model for predicting coronary heart disease: Enhancing clinical decision-making with machine learning

Lang Deng[1,2], Kongjie Lu[1,2]*, Huanhuan Hu[1,2]*

1 Huzhou Central Hospital, Fifth School of Clinical Medicine of Zhejiang Chinese Medical University, Huzhou, China, 2 Huzhou Central Hospital, Affiliated Central Hospital of Huzhou University, Huzhou, China

* karrymama@126.com (HH); lkj583@sina.com (KL)

## Abstract

### Background

Coronary Heart Disease (CHD) is one of the major burdens of cardiovascular diseases worldwide. Traditional diagnostic methods, such as coronary angiography and electrocardiogram, face challenges including high costs, subjectivity, and high misdiagnosis rates. To address these issues, this study proposes a prediction framework for CHD based on the LightGBM algorithm, aiming to improve the accuracy and interpretability of CHD risk prediction.

### Methods

This study utilized three publicly available datasets: BRFSS_2015, Framingham, and Z-Alizadeh Sani. The BRFSS_2015 dataset was used for model training, while the Framingham and Z-Alizadeh Sani datasets were employed for validation. Data preprocessing included cleaning, feature engineering, and handling missing values. The LightGBM model was selected for its efficiency and performance, and SHAP (SHapley Additive exPlanations) values were used to enhance model interpretability. Model performance was evaluated using metrics such as accuracy, precision, recall, F1-score, and AUROC. A CHD scoring system was developed based on the model's predictions to assist clinicians in risk assessment.

### Results

The LightGBM model demonstrated excellent performance, achieving an accuracy of 90.60% and an AUROC of 81.06% on the BRFSS_2015 dataset. After parameter tuning, the model's accuracy improved to 90.61%, and the AUROC increased to 81.11%. On the Framingham dataset, the accuracy improved from 83.96% to 85.26%, and the AUROC increased from 62.86% to 67.37%. On the Z-Alizadeh Sani dataset,

**Data availability statement:** Third-party data was publicly sourced for this study from the Behavioral Risk Factor Surveillance System (BRFSS) survey data for 2015 (https://www.kaggle.com/datasets/cdc/behavioral-risk-factor-surveillance-system). Third-party data was also publicly sourced for this study from the Framingham cohort dataset (Framingham Heart Study) (https://www.kaggle.com/datasets/noeyislearning/framingham-heart-study). Third-party data was also publicly sourced for this study from the Z-Alizadeh Sani dataset (classification of coronary artery disease) (https://www.kaggle.com/datasets/saeedeheydarian/classification-of-coronary-artery-disease). The authors confirm others can replicate the study findings in their entirety by directly obtaining data from the Behavioral Risk Factor Surveillance System (BRFSS) survey data for 2015, the Framingham cohort dataset (Framingham Heart Study), and the Z-Alizadeh Sani dataset (classification of coronary artery disease), and following information outlined in the Methods section. The authors had no special access privileges that others would not have when attempting to access the minimal data from the Behavioral Risk Factor Surveillance System (BRFSS) survey data for 2015, the Framingham cohort dataset (Framingham Heart Study), and the Z-Alizadeh Sani dataset (classification of coronary artery disease). All other relevant data for this study are publicly available from the GitHub repository (https://github.com/qwer1234784/LightGBM-Model.git).

**Funding:** The author(s) received no specific funding for this work.

**Competing interests:** The authors have declared that no competing interests exist.

the accuracy improved from 78.69% to 80.33%, and the precision increased from 74.40% to 76.36%.

## Conclusions

SHAP analysis revealed that age, smoking status, diabetes, hypertension, and high cholesterol were the most influential features in predicting CHD risk. The developed CHD scoring system provided a user-friendly tool for clinicians to assess patient risk levels effectively.

## Introduction

Coronary Heart Disease (CHD) remains one of the leading causes of mortality worldwide, posing a significant burden on global healthcare systems. According to the World Health Organization (WHO), cardiovascular diseases account for nearly 18 million deaths annually, with CHD being a major contributor [1]. Traditional diagnostic methods, such as coronary angiography and electrocardiograms, while effective, are often associated with high costs, subjectivity, and potential misdiagnosis rates [2]. Coronary angiography, considered the gold standard for CHD diagnosis, is invasive, expensive, and not universally accessible, particularly in low-resource settings [3]. Electrocardiograms, although less invasive, can be subjective and may fail to detect early-stage CHD, leading to delayed diagnosis and treatment [4]. These limitations have spurred the exploration of alternative approaches, particularly those leveraging machine learning (ML) techniques, to enhance the accuracy, efficiency, and accessibility of CHD risk prediction.

In recent years, ML algorithms have gained traction in the medical field due to their ability to process large datasets and identify complex patterns that may not be apparent through conventional methods. Various algorithms, including K-Nearest Neighbors (KNN) [5], Support Vector Machines (SVM) [6], Decision Trees [7], and XGBoost [8], have been employed to develop predictive models for CHD. These models have demonstrated promising results in identifying key risk factors such as age, hypertension, diabetes, and smoking status, which are critical in assessing an individual's likelihood of developing CHD. For instance, Souza et al. [9] utilized a KNN-based model to predict CHD risk, achieving high accuracy in identifying patients with early-stage disease. Similarly, SVM models have been effective in handling high-dimensional data, making them suitable for analyzing complex clinical datasets [10]. Decision Trees, known for their interpretability, have been widely used in clinical decision support systems to stratify patients based on their risk profiles [11]. XGBoost, a gradient boosting algorithm, has shown exceptional performance in handling imbalanced datasets, a common challenge in CHD prediction [12]. These advancements highlight the potential of ML in transforming CHD diagnosis and risk assessment, offering a more scalable and cost-effective alternative to traditional methods. However, Existing ML models for CHD prediction face challenges like limited dataset diversity, small sample sizes, and imbalanced data, which hinder generalizability and performance [13]. For

instance, Patel et al. [3] found varying model performance across ethnic groups, while Wahab Sait et al. [12] struggled with small datasets and lack of validation. Imbalanced data further reduces accuracy and recall [14], and complex feature interactions can lead to suboptimal predictions [11]. These issues highlight the need for robust, interpretable models that address dataset limitations and provide actionable clinical insights [10].

This study proposes a predictive framework for coronary heart disease (CHD) using the LightGBM algorithm, known for its efficiency and scalability in handling large, high-dimensional datasets [15]. To enhance model interpretability, SHapley Additive exPlanations (SHAP) values are incorporated, offering insights into feature contributions and improving transparency for clinical decision-making [16]. The primary goal is to develop a reliable and interpretable CHD prediction model validated across multiple datasets, including BRFSS_2015, Framingham, and Z-Alizadeh Sani, ensuring generalizability across diverse populations. Additionally, a user-friendly CHD scoring system is proposed to help healthcare providers assess patient risk and design personalized prevention and treatment plans. In summary, this study advances machine learning applications in healthcare by combining LightGBM with SHAP values to create a robust, interpretable CHD prediction model, aiming to improve early diagnosis, risk assessment, and patient outcomes while reducing the burden of cardiovascular diseases.

## Methods and materials

### Datasets acquisition

This study employed three publicly accessible datasets: BRFSS_2015, Framingham, and Z-Alizadeh Sani, obtained from Kaggle:

(1) BRFSS_2015: https://www.kaggle.com/datasets/cdc/behavioral-risk-factorsurveillance-system

(2) Framingham: https://www.kaggle.com/datasets/noeyislearning/framingham-heart-study

(3) Z-AlizadehSani: https://www.kaggle.com/datasets/saeedeheydarian/classification-of-coronary-artery-disease

The model was trained on the BRFSS_2015 dataset and validated using the Framingham and Z-Alizadeh Sani datasets. Data preprocessing was conducted to retain common features and derive additional relevant attributes. The BRFSS_2015 dataset consists of 253,680 samples, the Framingham dataset includes 4,240 samples, and the Z-Alizadeh Sani dataset contains 303 samples. Each dataset comprises 8 attributes, which are categorized as follows:

(1) Age: age of the individual in years.

(2) Sex: gender of the individual (0 = female, 1 = male).

(3) BMI: body Mass Index, a measure of body fat based on height and weight.

(4) Smoker: indicates whether the individual is a smoker (0 = no, 1 = yes).

(5) Diabetes: indicates the presence of diabetes (0 = no, 1 = yes).

(6) HighBP: indicates the presence of high blood pressure (0 = no, 1 = yes).

(7) HighChol: indicates the presence of high cholesterol (0 = no, 1 = yes).

(8) CHD Diagnosis: target attribute that indicates whether the individual has been diagnosed with coronary heart disease (0 = Normal, 1 = CHD).

Continuous variables (e.g., Age, BMI) were grouped by median values: "<median" and "≥ median." Categorical variables (e.g., Sex, Diabetes, HighBP, HighChol) were analyzed for group statistics, calculating the odds ratio (OR) and p-value.

The association between each variable and CHD was evaluated using 2x2 contingency tables with the formula (1):

$$OR = \frac{(a/c)}{(b/d)}$$

(1)

Here, a and b denote CHD patients in exposed and non-exposed groups, while c and d represent the normal group.

Fisher's exact test determined the p-value for significance, with p<0.05 indicating a significant association. Statistical analysis of the raw data (Tables 1–3) identified significant associations through p-value assessment.

## Data collection and preprocessing

In this study, we developed a comprehensive coronary heart disease (CHD) diagnostic framework, encompassing the entire process from data collection and preprocessing to feature engineering, model training, and the establishment of a CHD scoring system. This framework demonstrates our systematic approach and innovative exploration in the field of CHD prediction. The research began with data collection, where we utilized three publicly available datasets: BRFSS_2015, Framingham, and Z-Alizadeh Sani. The BRFSS_2015 dataset, comprising 253,680 samples, was used

**Table 1. Statistical analysis of characteristics of the BRFSS_2015 data set.**

| Characteristics | CDH (n = 23893) | Normal (n = 229787) | OR | P-value |
|---|---|---|---|---|
| **Age** | | | 0.747 | <0.001*** |
| < 55 | 8501 | 97644 | | |
| ≥ 55 | 15392 | 132143 | | |
| **Sex** | | | 0.555 | <0.001*** |
| Female | 10205 | 131769 | | |
| Male | 13688 | 98018 | | |
| **BMI** | | | 1.004 | 0.748 |
| < 27 | 10571 | 101412 | | |
| ≥ 27 | 13322 | 128375 | | |
| **Smoker** | | | 0.555 | <0.001*** |
| No | 10205 | 131769 | | |
| Yes | 13688 | 98018 | | |
| **Diabetes** | | | 0.285 | <0.001*** |
| No | 15351 | 198352 | | |
| Yes | 8542 | 31435 | | |
| **HighBP** | | | 0.218 | <0.001*** |
| No | 5965 | 138886 | | |
| Yes | 17928 | 90901 | | |
| **HighChol** | 7140 | | 0.279 | <0.001*** |
| No | 7140 | 138949 | | |
| Yes | 16753 | 90838 | | |
| **Stroke** | | | 0.144 | <0.001*** |
| No | 19956 | 223432 | | |
| Yes | 3937 | 6355 | | |
| **Obesity** | | | 0.67 | <0.001*** |
| No | 6537 | 82689 | | |
| Yes | 17356 | 147098 | | |

**Table 2. Statistical analysis of characteristics of the Framingham data set.**

| Characteristics | CDH (n = 644) | Normal (n = 3596) | OR | P-value |
|---|---|---|---|---|
| **Age** | | | 0.991 | 0.932 |
| < 49 | 315 | 1767 | | |
| ≥ 49 | 329 | 1829 | | |
| **Sex** | | | 0.612 | <0.001*** |
| Female | 301 | 2119 | | |
| Male | 343 | 1477 | | |
| **BMI** | | | 0.989 | 0.898 |
| < 25.4 | 320 | 1797 | | |
| ≥ 25.4 | 324 | 1799 | | |
| **Smoker** | | | 0.612 | <0.001*** |
| No | 301 | 2119 | | |
| Yes | 343 | 1477 | | |
| **Diabetes** | | | 0.295 | <0.001*** |
| No | 604 | 3527 | | |
| Yes | 40 | 69 | | |
| **HighBP** | | | 0.374 | <0.001*** |
| No | 319 | 2604 | | |
| Yes | 325 | 992 | | |
| **HighChol** | | | 0.695 | <0.001*** |
| No | 322 | 2122 | | |
| Yes | 322 | 1474 | | |
| **Stroke** | | | 0.225 | <0.001*** |
| No | 633 | 3582 | | |
| Yes | 11 | 14 | | |
| **Obesity** | | | 0.684 | <0.001*** |
| No | 245 | 1701 | | |
| Yes | 399 | 1895 | | |

for model training, while the Framingham (n = 4,240) and Z-Alizadeh Sani (n = 303) datasets were employed for model validation.

During the data preprocessing phase, we conducted thorough data cleaning to ensure data quality and consistency. We standardized and unified the column names across all datasets to maintain consistent feature naming. Key features such as age, gender, BMI, smoking status, diabetes, hypertension, high cholesterol, and CHD were identified and harmonized across datasets.

For the BRFSS_2015 dataset, no missing values were detected, and thus no imputation was performed. Similarly, the Z-Alizadeh Sani dataset was found to be complete with no missing values. For the Framingham dataset, missing BMI values were imputed using the mean value to ensure data integrity.

We addressed missing and outlier values to enhance data reliability. In In the Framingham dataset, missing BMI values were estimated using mean values to ensure data integrity. Categorical variables were coded as numeric data; for example, sex was coded as 0 for female and 1 for male; smoking status was coded as 0 for female and 1 for male; smoking status was coded as 0 for nonsmokers and 1 for smokers. No missing values were found in the BRFSS_2015 dataset. The binary variables HighBP (high blood pressure), HighChol (high cholesterol), CholCheck, Smoker, and Stroke represent the individual's health behavior and chronic disease status. These characteristics were coded as 0/1, where 0

**Table 3. Statistical analysis of characteristics of the Z-AlizadehSani data set.**

| Characteristics | CDH (n = 216) | Normal (n = 87) | OR | P-value |
|---|---|---|---|---|
| **Age** | | | 1.071 | 0.801 |
| < 58 | 108 | 42 | | |
| ≥ 58 | 108 | 45 | | |
| **Sex** | | | 0.742 | 0.25 |
| Female | 86 | 41 | | |
| Male | 130 | 46 | | |
| **BMI** | | | 1.023 | <0.001*** |
| < 26.8 | 108 | 43 | | |
| ≥ 26.8 | 108 | 44 | | |
| **Smoker** | | | 0.742 | 0.25 |
| No | 86 | 41 | | |
| Yes | 130 | 46 | | |
| **Diabetes** | | | 0.221 | <0.001*** |
| No | 136 | 77 | | |
| Yes | 80 | 10 | | |
| **HighBP** | | | 0.273 | <0.001*** |
| No | 69 | 55 | | |
| Yes | 147 | 32 | | |
| **HighChol** | | | 1.06 | 0.896 |
| No | 137 | 54 | | |
| Yes | 79 | 33 | | |
| **Stroke** | | | 0.616 | <0.001*** |
| No | 212 | 86 | | |
| Yes | 4 | 1 | | |
| **Obesity** | | | 1.055 | 0.891 |
| No | 67 | 26 | | |
| Yes | 149 | 61 | | |

indicates "no" or "did not occur" and 1 indicates "yes" or "occurred". ". No missing values were detected in the Z-Alizadeh Sani dataset. Since all variables were fully available during preprocessing for both the Z-Alizadeh-S and BRFSS_2015 datasets, there was no need for missing value imputation or sample exclusion. Data integrity was validated beforehand to ensure the quality of both datasets.

In the Z-Alizadeh Sani coronary artery disease dataset, a total of 303 samples were included, of which 216 patients with coronary artery disease accounted for approximately 71% of the total sample size, and 87 patients with non-coronary artery disease accounted for approximately 29%. Although this dataset shows an overall distribution of patients with coronary artery disease in the majority, in the actual training process, the model may still have insufficient discriminative ability for a few categories due to the difference in the proportion of the sample size. In order to improve the model's ability to recognize various categories, especially to accurately discriminate the minority category, this paper applies the SMOTE method to oversample the minority category samples in the training set. It avoids the risk of overfitting caused by directly copying the original samples. After SMOTE, the number of categories in the training set is adjusted to: CAD = 1: 216 samples, CAD = 0: 216 samples. This ensures an equal number of samples in the two categories, so that the model will not be biased towards the majority class when learning discriminative features, and improves the prediction ability for the minority class samples. It should be noted that SMOTE is only applied on the training set, and the test set retains the original distribution to ensure the objectivity and generalization of the model evaluation.

Considering the imbalance between CHD patients (216 cases) and non-CHD patients (87 cases) in the Z-Alizadeh Sani dataset, this study applied the SMOTE (Synthetic Minority Over-sampling Technique) method to oversample the minority class. This approach was chosen to prevent the model from being biased towards the majority class. Other techniques such as class weighting and cost-sensitive learning were also explored, but SMOTE was selected for its ability to increase the diversity of minority class samples without causing overfitting.

In the feature selection stage, we employed statistical analysis methods, calculating odds ratios (OR) and p-values to evaluate the association between features and CHD. These preprocessing steps ensured data integrity and consistency, laying a solid foundation for subsequent feature engineering and model training.

## Feature engineering and model training

During the feature engineering phase, we incorporated SHapley Additive exPlanations (SHAP) values to enhance model interpretability. SHAP values allowed us to quantify the contribution of each feature to the model's predictions, providing a scientific basis for feature importance. This approach not only clarified the model's decision-making process but also offered clinicians intuitive and reliable explanations, facilitating the model's application in real-world medical scenarios. In the model comparison phase, we conducted a comprehensive evaluation of various advanced machine learning algorithms to identify the most effective approach for CHD prediction. The algorithms tested included XGBoost [8], Random Forest [17], Bagging [18], CatBoost [19],Decision Tree [20], and K-Nearest Neighbors [21]. Each model was rigorously tested and fine-tuned by adjusting hyperparameters to optimize prediction accuracy, robustness, and generalizability. This step was critical to ensure that the selected model could handle the complexities of CHD prediction, including high-dimensional data, imbalanced class distributions, and diverse patient populations.

To provide a holistic evaluation of the models, we employed radar charts as a visualization tool. These charts allowed us to compare the performance of each model across multiple key metrics, including accuracy, precision, recall, F1-score, and the area under the receiver operating characteristic curve (AUROC). By analyzing these metrics, we were able to assess not only the predictive power of each model but also its stability and reliability in real-world clinical scenarios. For instance, while one model might excel in accuracy, another might demonstrate superior recall, which is crucial for identifying true positive CHD cases. This multi-metric approach ensured that the final model selection was well-rounded and capable of meeting the diverse needs of healthcare providers.

## CHD scoring system development

Based on the predictions generated by the best-performing model, we developed a user-friendly CHD scoring system. This system was designed to assist healthcare providers in quickly and accurately assessing a patient's risk of developing CHD. The scoring system features an intuitive user interface where clinicians can input patient data, such as demographic information, medical history, and clinical measurements. The system then processes this data and generates a CHD risk score, which can be used to guide clinical decision-making. By providing a quantifiable risk assessment, the scoring system helps clinicians identify high-risk patients earlier, enabling timely interventions and personalized prevention strategies. This not only reduces the risk of disease progression but also lowers healthcare costs by minimizing unnecessary tests and treatments.

The integration of this scoring system into clinical practice has the potential to significantly enhance the efficiency and effectiveness of CHD screening. For example, in primary care settings, the system can be used to prioritize patients for further diagnostic testing or specialist referrals. In hospital settings, it can support the development of tailored treatment plans for patients with varying risk levels. Moreover, the system's interpretability, facilitated by SHAP values, ensures that clinicians can understand the factors contributing to each patient's risk score, fostering trust and adoption of the tool.

The overall workflow of the CHD diagnosis process is illustrated in **Fig 1**. The process begins with data collection and preprocessing, followed by model training and evaluation. Once the optimal model is selected, it is integrated into the

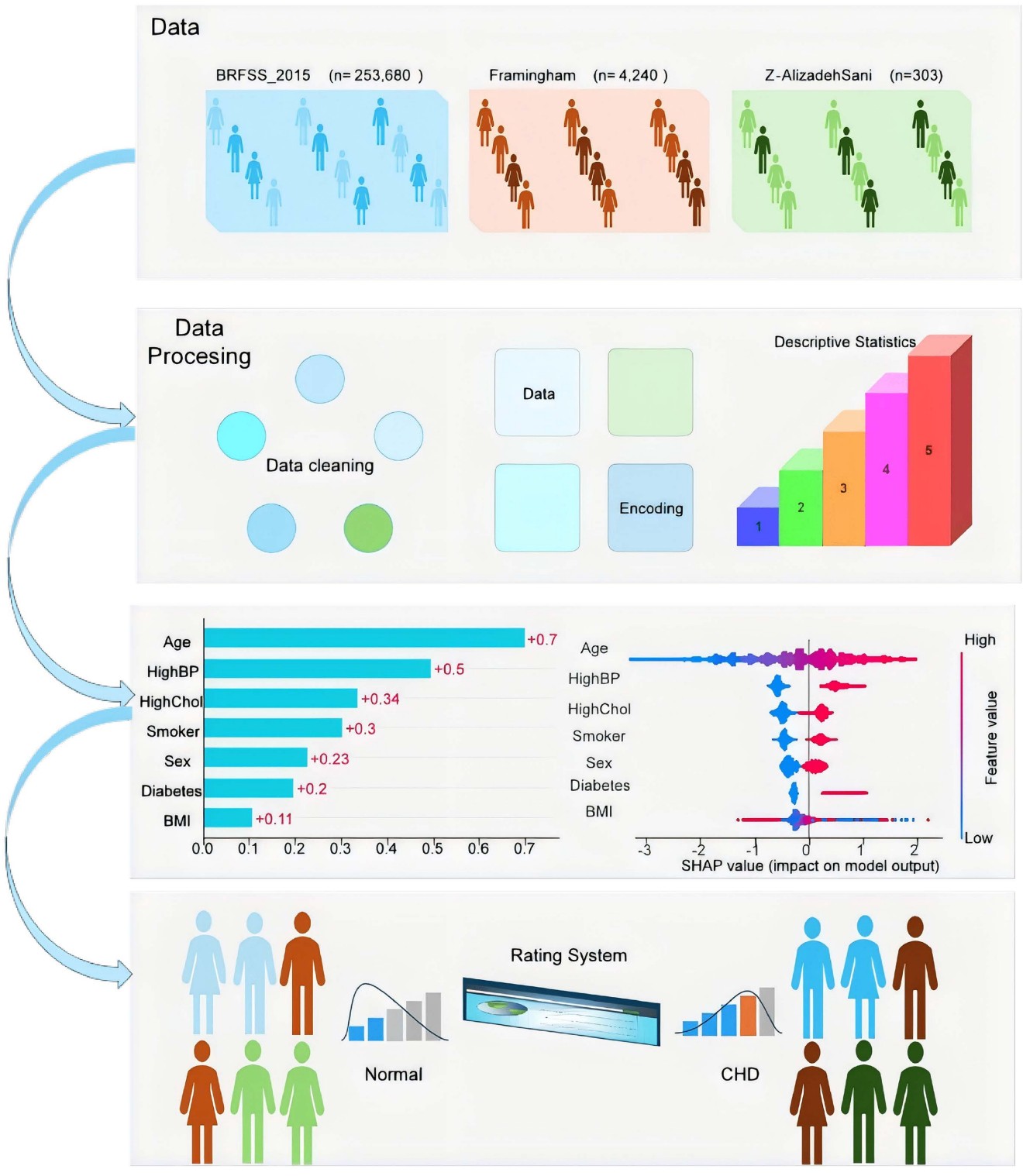

**Fig 1. Flow chart of coronary heart disease diagnosis.**

CHD scoring system, which is then deployed for clinical use. This end-to-end framework not only leverages advanced machine learning techniques but also emphasizes practicality and interpretability, ensuring that the solution is both scientifically robust and clinically applicable.

In summary, this study not only advances the field of machine learning in healthcare by demonstrating the effectiveness of LightGBM and other algorithms in CHD prediction but also bridges the gap between research and clinical practice. By developing a reliable, interpretable, and user-friendly CHD scoring system, we aim to empower healthcare providers with actionable insights, ultimately improving patient outcomes and reducing the global burden of cardiovascular diseases.

## Evaluating indicator

Several evaluation metrics were used in this study to comprehensively measure the performance of the algorithm in predicting coronary heart disease, including Accuracy, Precision, Recall, F1 Score, Area Under the ROC Curve (AUROC), and Precision-Recall (PR) curve. While accuracy is a commonly used metric, it can be misleading in imbalanced datasets. Therefore, we emphasize the importance of AUROC, F1-score, and precision-recall curves, which provide a more comprehensive assessment of model performance, especially in clinical contexts where false positives and false negatives have significant consequences.

These metrics are defined as follows (see equations (2)-(5)):
True positive (TP): number of samples correctly predicted by the model to be coronary heart disease;

True negative (TN): number of samples correctly predicted as normal by the model;

False positives (FP): normal samples incorrectly predicted by the model as having coronary heart disease;

False negative (FN): samples with coronary heart disease that the model incorrectly predicted as normal.

Among them, Accuracy (Acc) indicates the proportion of correctly classified samples, Precision (Pre) indicates the proportion of samples predicted to be positive that are actually positive, Recall (Recall) indicates the proportion of all positive samples that are correctly predicted, and the F1 score is the harmonic mean of Precision and Recall.AUROC (i.e., area under the ROC curve) reflects the overall discriminatory ability of the model for positive and negative samples, and the closer the value is to 1, the higher the value will be. AUROC (i.e., the area under the ROC curve) reflects the model's ability to distinguish between positive and negative samples, and the closer the value is to 1, the better the model performance.

Accuracy (Acc) is defined as the ratio of correctly classified samples to the total number of samples in the test set, as shown in Equation (2).

$$Acc = \frac{TP + TN}{TP + TN + FP + FN} \tag{2}$$

Precision (Pre) refers to the ratio of the number of true positive samples to the number of samples predicted as positive in the test set, as shown in Equation (3).

$$Pre = \frac{TP}{TP + FP} \tag{3}$$

Recall refers to the ratio of the number of correctly predicted positive samples to the number of true positive samples in the test set, as shown in Equation (4).

$$Recall = \frac{TP}{TP + FN} \tag{4}$$

F1-score is the harmonic mean of Precision and Recall, as shown in Equation (5).

$$\text{F1} - \text{score} = 2 \times \frac{\text{Pre} \times \text{Recall}}{\text{Pre+Recall}}$$

(5)

In addition, considering that some datasets have significant category imbalance, this paper introduces the Precision-Recall curve as a supplementary assessment tool. Compared with the ROC curve, the PR curve is more valuable when the minority samples are the focus of attention, and can more effectively reveal the performance of the model in identifying patients with coronary heart disease.

Through the comprehensive analysis of the above evaluation indexes, the generalization ability, robustness, and identification of key clinical risks of the model on different datasets can be evaluated from different dimensions, thus providing theoretical support for the design of the subsequent scoring system and model selection.

In order to provide clinical workers with an efficient and accurate risk assessment tool for CHD, this study developed a user-friendly scoring system based on advanced machine learning prediction models. At the core of the system is formula (6), which converts complex model output into understandable scores in a concise and clear way. Formula (6) is a common scoring formula.

$$\text{score} = \sum_{i=1}^{j} \alpha_i w_i$$

(6)

Here, $j$ denotes the total number of features included in the model. Each $\alpha_i$ is the coefficient learned during training, indicating how strongly feature $i$ contributes to CHD risk prediction. Each $w_i$ is the actual value of feature $i$ (for example, a patient's age, sex, cholesterol level, or other clinical measurement).

## Results

### Model performance assessment

We first benchmarked six classifiers—XGBoost [8], Random Forest, Bagging [18], CatBoost [19],Decision Tree [20], and K-Nearest Neighbors [21]—on the BRFSS_2015 training set using five-fold cross-validation to compute accuracy, precision, recall, F1-score, and AUROC. The accuracy of these models generally reached about 90%, showing strong classification performance. Among them, LightGBM performed best in the two key indicators of accuracy (90.60%) and AUROC (81.06%). Although other models also performed well in some specific indicators, LightGBM was more balanced and superior in overall performance. In order to more intuitively show the performance differences between different models, we drew a radar chart (Fig 2). It not only clearly shows the specific performance of each model in each indicator, but also intuitively reflects the stable advantages of the LightGBM model in multiple dimensions. It shows the performance of each model under the original features of the BRFSS_2015 dataset, including accuracy, precision, recall, F1 score, and AUROC. Through the radar chart, we can clearly see that LightGBM outperforms other models in multiple indicators, especially in AUROC and F1 score, where LightGBM has more obvious advantages.

After selecting LightGBM, we carefully tuned its parameters to further improve the model performance. The tuning process mainly revolves around key parameters such as learning rate (learning_rate), number of trees (n_estimators), maximum depth (max_depth), and number of leaf nodes (num_leaves). In order to ensure the generalization ability of the model, we used five-fold cross validation to evaluate the effect of different parameter combinations. During the parameter tuning process, we used the grid search method to systematically traverse multiple parameter combinations to find the optimal model configuration. Grid search uses an exhaustive method to find the best combination in a predefined parameter space. Although the computational cost is high, it can ensure that the global optimal solution is found. After adjusting the parameters, the performance of the LightGBM model on multiple datasets has been significantly improved. Specifically, on the BRFSS_2015 dataset, the accuracy of the model increased from 90.60% to 90.61%, the precision increased

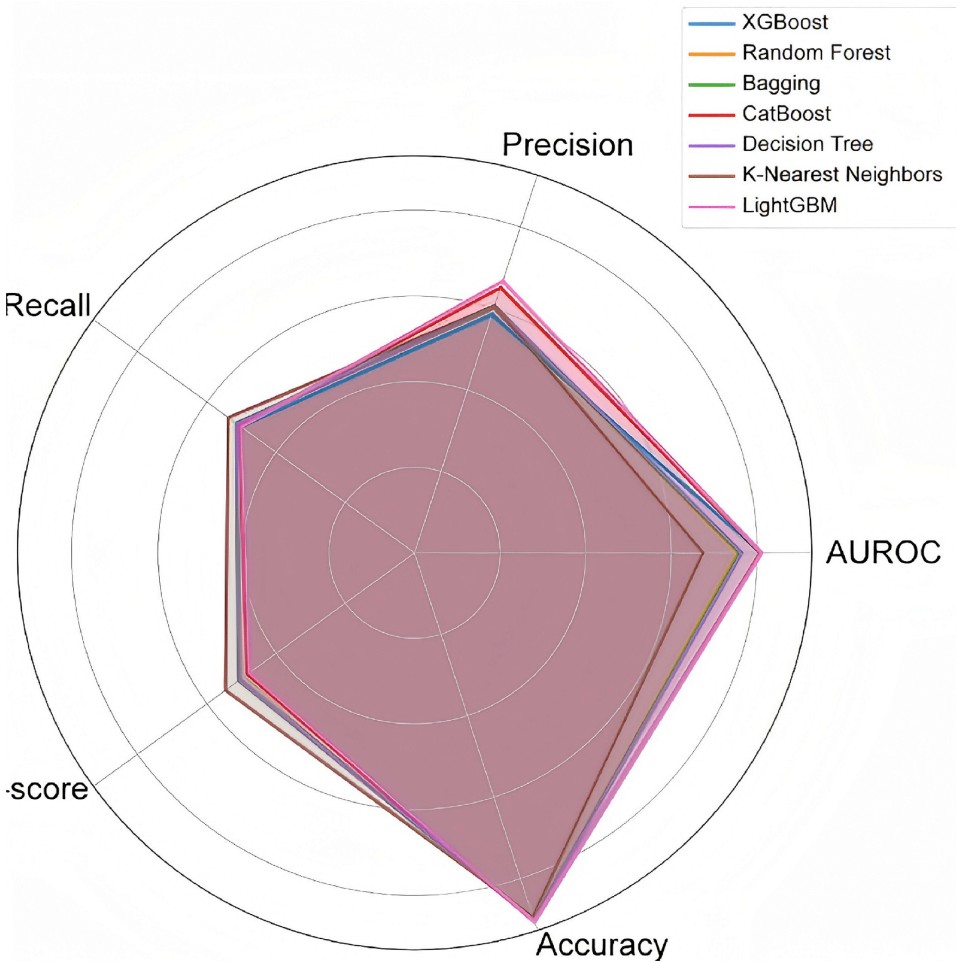

**Fig 2. Comparison of the performance of the different models in the BRFSS _ 2015 data set.**

significantly from 66.73% to 73.89%, and the AUROC also increased from 81.06% to 81.11%. On the Framingham data-set, the accuracy increased from 83.96% to 85.26%, and the AUROC increased from 62.86% to 67.37%. On the Z-AlizadehSani dataset, the accuracy increased from 78.69% to 80.33%, and the precision increased from 74.40% to 76.36%. These improvements show that through careful parameter tuning, the performance of the LightGBM model on different datasets has been significantly improved, especially in terms of precision and AUROC (Fig 3).

In addition, in datasets with more significant category imbalance (e.g., Z-Alizadeh Sani and Framingham), relying on Accuracy and AUROC alone does not fully reflect the model's performance in real-world prediction tasks. Especially in clinical scenarios such as coronary heart disease prediction, the ability to recognize positive cases (diseased individuals) is even more important. Therefore, this paper further focuses on the metrics of Precision, Recall and F1-score. Compared with Precision, F1-score can measure both the accuracy and completeness of the model in identifying positive samples, which is especially suitable for tasks such as coronary heart disease prediction, where the number of "positive" samples is small, but identifying them is of high clinical value.

The tabular results show that the parameter optimization improves the model performance differently in the three datasets, especially in the Z-Alizadeh Sani dataset. In this dataset, the model's accuracy increased from 78.6% to 80.3%,

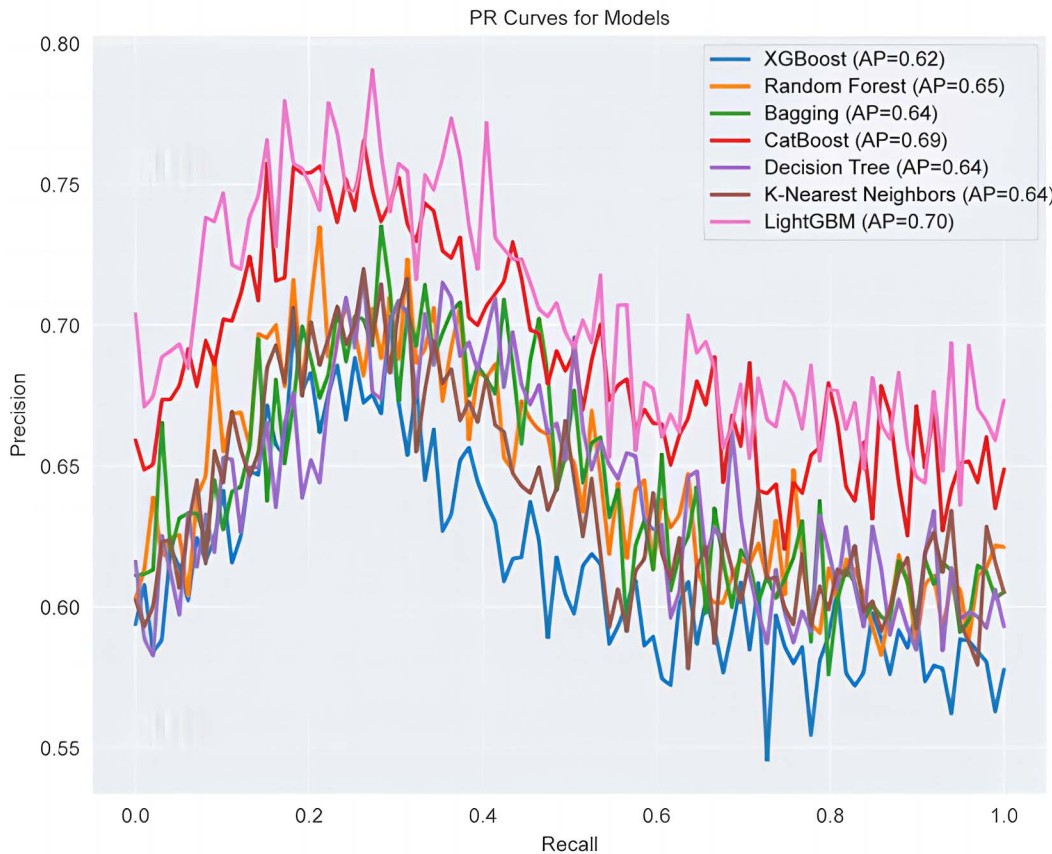

**Fig 3. Comparison of PR plots of different models.**

precision increased from 74.4% to 76.3%; more critically, the recall increased from 73.58% to 76.36%, and the F1-score also increased from 73.9% to 76.36%. This indicates that the parameter-adjusted model can more effectively identify CHD-positive patients with higher sensitivity and stability, which significantly enhances the practical value of the model in real clinical scenarios.

In contrast, in the Framingham dataset, although the AUROC increased from 62.86% to 67.37%, the F1-score decreased from 50.8% to 49.7%, and the improvement of precision and recall was smaller. This phenomenon may be related to the relatively small sample size and lack of variable diversity in this dataset, and also suggests that the generalization ability of the model is limited in this kind of single group.

In the BRFSS_2015 dataset, on the other hand, although the accuracy remains almost unchanged (90.6%) after parameter adjustment, the precision rate improves from 66.7% to 73.8%, suggesting that the model predictions are more reliable, although the improvements in the recall rate and F1-score are relatively limited.

In summary, the results of this paper show that although the parameter tuning has different degrees of influence on different datasets, the most significant comprehensive performance improvement is obtained on the Z-Alizadeh Sani dataset, especially in the recall and F1-score metrics. This further validates that relying only on "overall" evaluation criteria such as accuracy or AUROC may underestimate the usefulness of the model in medical classification tasks, while evaluating the F1-score and recall as key metrics more truly reflects the model's ability to discriminate key clinical populations.

As shown in Tables 4 and 5, we observe the performance comparison of different models on the BRFSS_2015 dataset, as well as the performance improvements of the LightGBM model before and after parameter tuning across multiple

**Table 4. Performance comparison of different models of the BRFSS_2015 data set.**

| Model | Acc (%) | Pre (%) | Recall (%) | F1-score (%) | AUROC (%) | Time(s) | Memroy Used(MB) |
|---|---|---|---|---|---|---|---|
| XGBoost | 90.56 | 58.4 | 50.08 | 47.75 | 80.79 | 1.9768 | 3.25 |
| Random Forest | 90.19 | 60.54 | 51.28 | 50.45 | 75.47 | 4.2968 | 66.4023 |
| Bagging | 90.13 | 60.34 | 51.42 | 50.73 | 75.93 | 1.1 | 7.2539 |
| CatBoost | 90.57 | 65.09 | 50.3 | 48.22 | 80.81 | 9.8932 | 12.5117 |
| Decision Tree | 90.19 | 60.54 | 51.28 | 50.45 | 76.47 | **0.1757** | 2.1367 |
| K-NearestNeighbors | 89.36 | 60.6 | **53.73** | **54.58** | 67.35 | 2.5791 | 13.9414 |
| LightGBM | **90.6** | **66.73** | 50.03 | 47.6 | **81.06** | 0.4426 | **1.7656** |

**Table 5. Performance comparison of the same models of the different data set.**

| Before and after parameter adjustment | Data set | Model | Acc (%) | Pre (%) | Recall (%) | F1-score (%) | AUROC (%) |
|---|---|---|---|---|---|---|---|
| Original parameters | BRFSS_2015 | LightGBM | 90.6 | 66.73 | 50.03 | 47.6 | 81.06 |
| | Framingham | LightGBM | 83.96 | 56.77 | **51.8** | **50.86** | 62.86 |
| | Z-AlizadehSani | LightGBM | 78.69 | 74.4 | 73.58 | 73.96 | 78.42 |
| n_estimators = 60, learning_rate = 0.2, num_leaves = 10, max_depth = 4, min_child_samples = 20, min_child_weight = 0.001, subsample = 1, colsample_bytree = 1 | BRFSS_2015 | LightGBM | **90.61** | **73.89** | 50.23 | 48.04 | **81.11** |
| | Framingham | LightGBM | **85.26** | **63.78** | 51.55 | 49.7 | **67.37** |
| | Z-AlizadehSani | LightGBM | **80.33** | **76.36** | **76.36** | **76.36** | 78.55 |

datasets. Fig 3 reflects the reference value when a small number of class samples are the focus of attention, which can more effectively reveal the performance of the model in identifying patients with coronary heart disease. We further visualize these performance differences and the effects of parameter adjustments, making the results more intuitive and easier to interpret (Fig 3). Together, these charts and data support our conclusion that LightGBM can significantly enhance prediction performance after proper tuning. In the model selection process, we not only considered traditional machine learning models but also incorporated ensemble learning methods that have demonstrated strong performance in recent years, such as XGBoost and CatBoost. These models excel in handling high-dimensional data and addressing class imbalance issues, particularly in medical datasets, where they can effectively capture complex nonlinear relationships. Additionally, we compared the computational efficiency of different models and found that LightGBM has significant advantages in terms of training speed and memory usage, which is especially critical for processing large-scale datasets.During the parameter tuning process, we employed a combination of grid search to improve the efficiency of parameter selection. Furthermore, we implemented an early stopping strategy to halt training when model performance no longer improved significantly, thereby preventing over-fitting. Each LightGBM parameter plays a critical role in optimizing model performance: n_estimators controls the number of trees, balancing accuracy and computational cost; learning_rate adjusts the step size during training, with smaller values improving generalization; num_leaves and max_depth manage tree complexity, where higher values capture detailed patterns but risk over-fitting; min_child_samples and min_child_weight ensure robust leaf nodes by setting minimum data or weight thresholds; and subsample and colsample_bytree reduce overfitting by limiting the samples or features used per tree. Together, these parameters fine-tune the model's accuracy, complexity, and generalization capabilities. For parameter tuning, we explored a wide range of hyper-parameters, including:

(1) n_estimators: [50, 100, 150, 200, 250, 300, 350, 400, 450, 500, 550]

(2) learning_rate: [0.01, 0.05, 0.1, 0.2, 0.3, 0.4, 0.5, 0.6, 0.7, 0.8, 0.9, 1.0]

(3) num_leaves: [31, 50, 70, 100, 150, 200, 250, 300, 350, 400, 450, 500]

(4) max_depth: [−1, 1, 3, 5, 7, 9, 11, 13, 15, 17, 19, 21, 23, 25]

(5) min_child_samples: [1, 2, 4, 8, 16, 20, 32, 64, 128, 256, 512, 1024]

(6) min_child_weight: [0.001, 0.01, 0.1, 1, 10, 100, 1000, 10000, 100000]

(7) subsample: [0.1, 0.2, 0.3, 0.4, 0.5, 0.6, 0.7, 0.8, 0.9, 1.0]

(8) colsample_bytree: [0.1, 0.2, 0.3, 0.4, 0.5, 0.6, 0.7, 0.8, 0.9, 1.0]

After extensive experimentation and optimization, the final selected parameters for the LightGBM model were carefully chosen to achieve a balance between predictive accuracy, computational efficiency, and generalization performance. These parameters were determined through a combination of grid search, ensuring that the model was fine-tuned to handle the complexities of the dataset while avoiding over-fitting. The final configuration is as follows:

(1) n_estimators: 60

(2) learning_rate: 0.2

(3) num_leaves: 10

(4) max_depth: 4

(5) min_child_samples: 20

(6) min_child_weight: 0.001

(7) subsample: 1

(8) colsample_bytree: 1

These optimized parameters were chosen to strike a balance between model complexity and generalization performance. For instance, a smaller num_leaves and max_depth were selected to prevent overfitting, while a moderate learning_rate ensured stable convergence during training. The subsample and colsample_bytree values of 1 indicate that the model utilizes the full dataset and all features, which was found to be optimal for this specific task.

In different clinical subgroups, we evaluated the model's predictive performance across gender, age, and diabetes status. The results indicated that the model performed better in the male population compared to the female population. In males, the model's AUC was 0.85, recall was 0.78, precision was 0.62, and F1 score was 0.69. In females, the AUC was 0.79, recall was 0.64, precision was 0.56, and F1 score was 0.59. This suggests that the model has a stronger ability to identify male patients, particularly in recall and overall discrimination ability (AUC), which were higher in males by approximately 0.06 to 0.09.

In the age subgroup, individuals aged below 50 years showed a slight advantage in predictive performance compared to those aged 50 years and above. For those under 50 years, the model's AUC was 0.83, recall was 0.72, precision was 0.59, and F1 score was 0.65. For those aged 50 years and above, the AUC was 0.81, recall was 0.68, precision was 0.58, and F1 score was 0.62. Despite the higher cardiovascular risk in older individuals, the model performed slightly better in the younger group, possibly due to more distinct clinical features in the younger population that helped differentiate the disease.

In terms of diabetes status, the model's predictive performance was significantly better in diabetic patients compared to non-diabetic individuals. In the diabetes subgroup, the AUC reached 0.86, recall was 0.76, precision was 0.63, and F1 score was 0.69. In the non-diabetic subgroup, the AUC was 0.80, recall was 0.65, precision was 0.55, and F1 score was 0.59. This reflects the model's greater sensitivity to identifying risks in diabetic patients, possibly due to clearer metabolic and physiological abnormalities in this group, making it easier for the model to detect early disease signals.

Overall, our tuned model for CHD prediction demonstrates excellent performance in terms of accuracy, stability, and generalization ability. It not only enhances the precision of CHD risk assessment but also serves as a reliable and efficient decision-support tool for clinical diagnosis by leveraging advanced machine learning techniques. While the model performs robustly overall, our subgroup analysis revealed certain performance differences—particularly with regard to gender and diabetes status—highlighting the importance of considering potential group-specific variations during model deployment. These findings suggest that further refinement may be necessary to ensure fairness and reliability across diverse populations. Looking ahead, we hope this study will offer strong support for CHD diagnosis and contribute valuable insights into the broader application of machine learning in the medical field. This research presents both theoretical innovations and promising practical value, offering new tools and perspectives for the early detection and treatment of heart disease Fig 4.

## Interpretability analysis

To deepen interpretability, we employed SHAP value visualizations across several plot types. First, waterfall charts depict the cumulative influence of each predictor on the risk score: in normal samples, age, BMI, smoking status, diabetes, and hypertension contribute negative SHAP values—pushing the predicted risk downward—whereas in CHD patients the same features, especially diabetes and high cholesterol, exert strong positive contributions toward risk (Fig 5A, 5B). Decision plots then trace the model's step-by-step reasoning, showing how incremental changes in key variables like age and smoking status drive the output trajectory for both groups (Fig 5C, 5D).

At the individual-sample level, force plots break down the exact "push" and "pull" effects of each feature: for a representative normal case, age and smoking pull the risk below the baseline, while in a CHD patient those features push the score above baseline, underscoring their pivotal roles in a patient's risk profile (Fig 5E, 5F). A combined feature-contribution plot contrasts all predictors between normal and CHD cohorts, highlighting which variables most distinctly separate the two populations (Fig 5G).

To explore how SHAP values vary across the feature spectrum, dependence plots reveal that older age steadily increases predicted CHD risk and that switching from non-smoker to smoker produces a marked rise in risk (Fig 6).

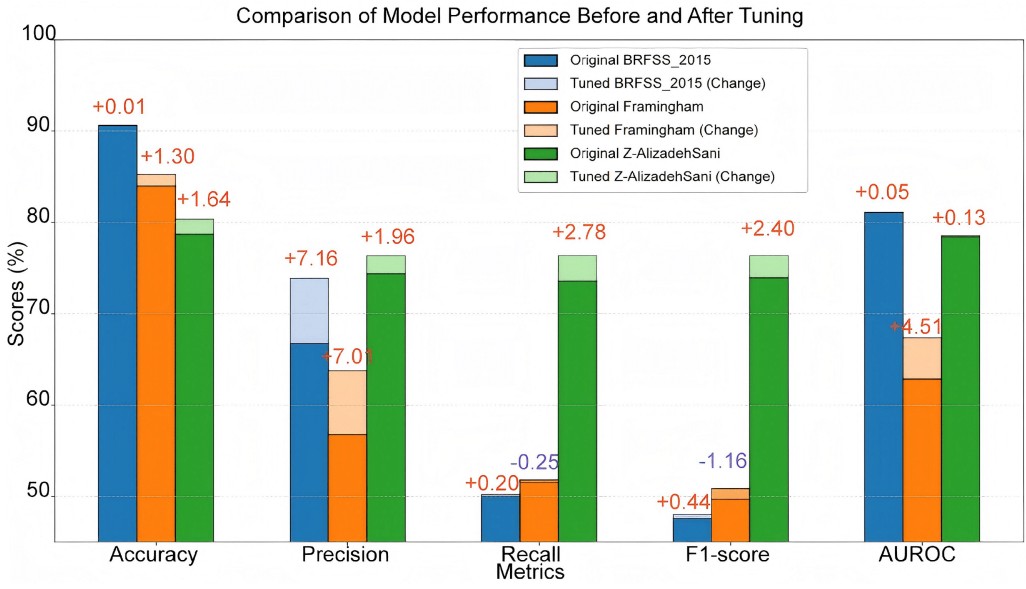

**Fig 4. Comaprison of model performance before and after tuning.**

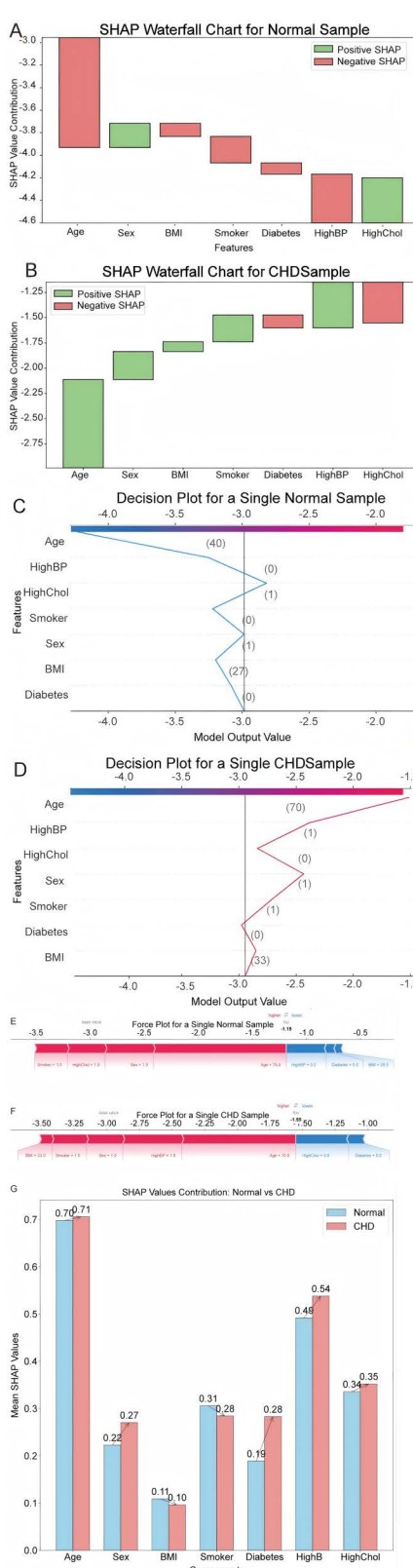

**Fig 5. A. Waterfall plot of normal samples; B: Waterfall plot of CHD patients; C: Decision diagram of normal samples; D: Decision diagram of CHD samples; E: Force diagram of normal samples; F: Force diagram of CHD patients; G: Feature contribution diagram of normal samples vs. CHD patients.**

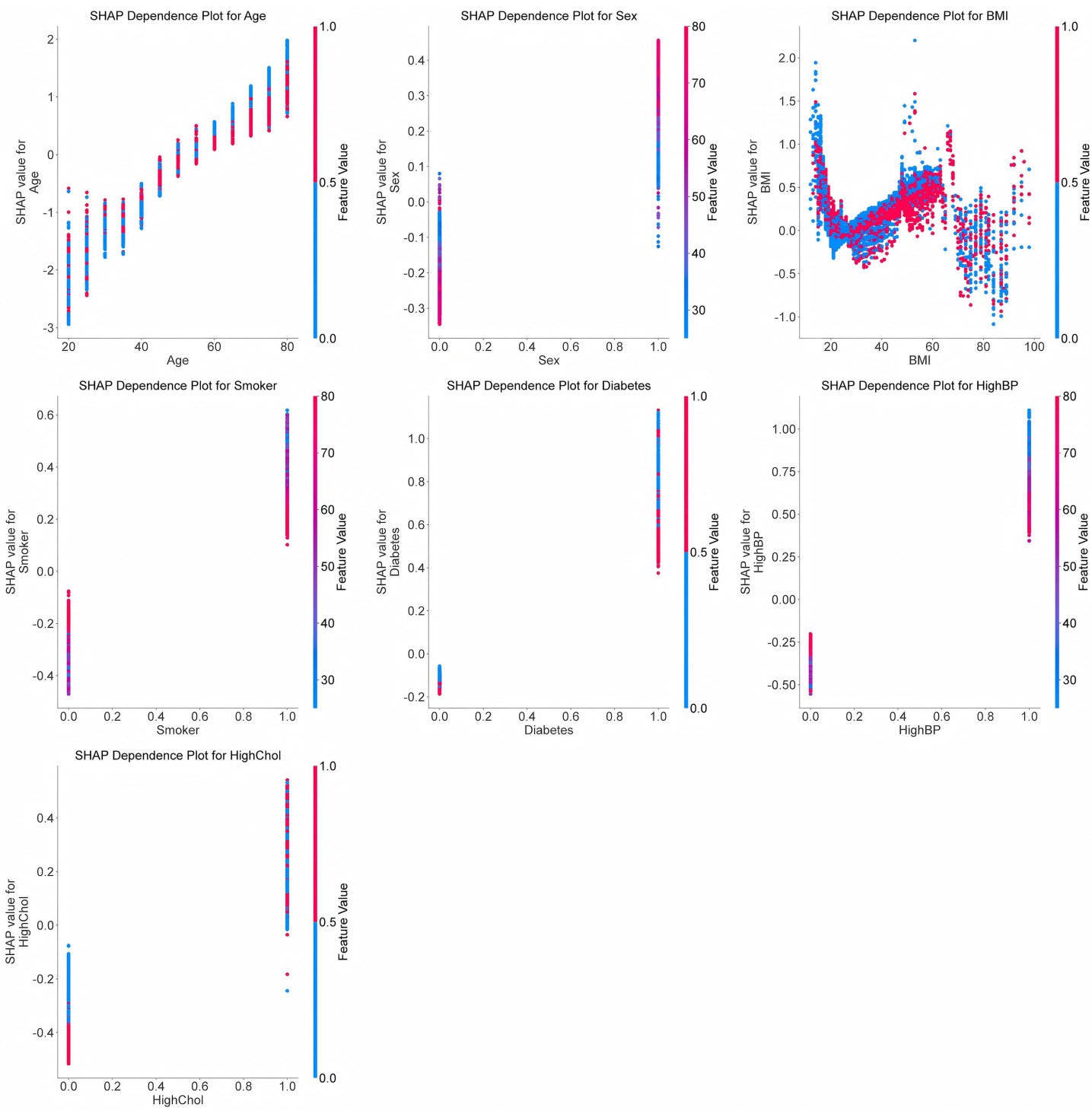

**Fig 6. The dependency plot for each feature.**

Finally, a SHAP summary plot aggregates each feature's average absolute impact on model outputs, ranking age, hypertension, high cholesterol, smoking, sex, diabetes, and BMI as the top drivers—and indicating whether each drives risk up or down (Fig 7).

To further validate the interpretability of the model and to add multiple perspectives to the validation, this paper introduces the LIME method. LIME is a local model interpretability method that helps us understand the decision logic of the model for a given sample by weighted sampling and fitting a linear model to each single prediction point. LIME is a local model interpretability method that helps us understand the decision logic of the model for a given sample by weighting each single prediction point and fitting a linear model. We compared the results of LIME with those of SHAP and found that they showed a high degree of consistency in the contribution of most features. In particular, both LIME and SHAP provided relatively consistent interpretations of age and smoking status (Smoker), demonstrating the importance of these two features for the prediction of coronary heart disease. In particular, age ranked high in both the LIME and SHAP analyses, demonstrating its high impact on prediction. In this way, we can not only see the global feature importance, but also further confirm the local feature contribution on specific samples through the LIME analysis. The comparison of the LIME and SHAP methods provides a more comprehensive support for the reliability of the model.

Fig 8 provides a comparison of the importance of the features of the coronary heart disease prediction model under the LIME method on different datasets. In the three datasets of BRFSS, Framingham, and Z-Alizadeh Sani, the important features revealed by LIME such as Age and HighBP were highly consistent with the SHAP analysis, especially in terms of stability across multiple models and sample intervals. The results are highly consistent with the SHAP analysis. Therefore, it can be assumed that the feature interpretations given by SHAP have a high degree of confidence, while the results of the LIME analysis serve as an effective corroborative support.

In order to verify the cross-data set stability of the models, this study analyzed the feature importance consistency of SHAP and LIME on different data sets. The results of the analysis showed that the importance rankings of features such as age, smoking status (Smoker) and high blood pressure (HighBP) remained stable across datasets such as BRFSS 2015, Framingham and Z-Alizadeh Sani. This result suggests that the model's dependencies on these features are highly stable and generalizable, not only to different sample populations, but also highly transferable across datasets. In addition,

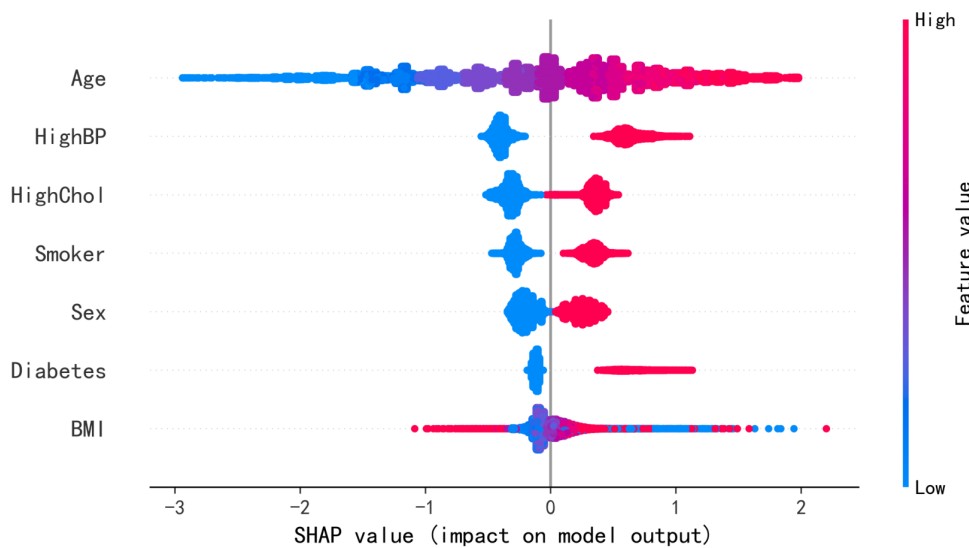

**Fig 7. SHAP summary.**

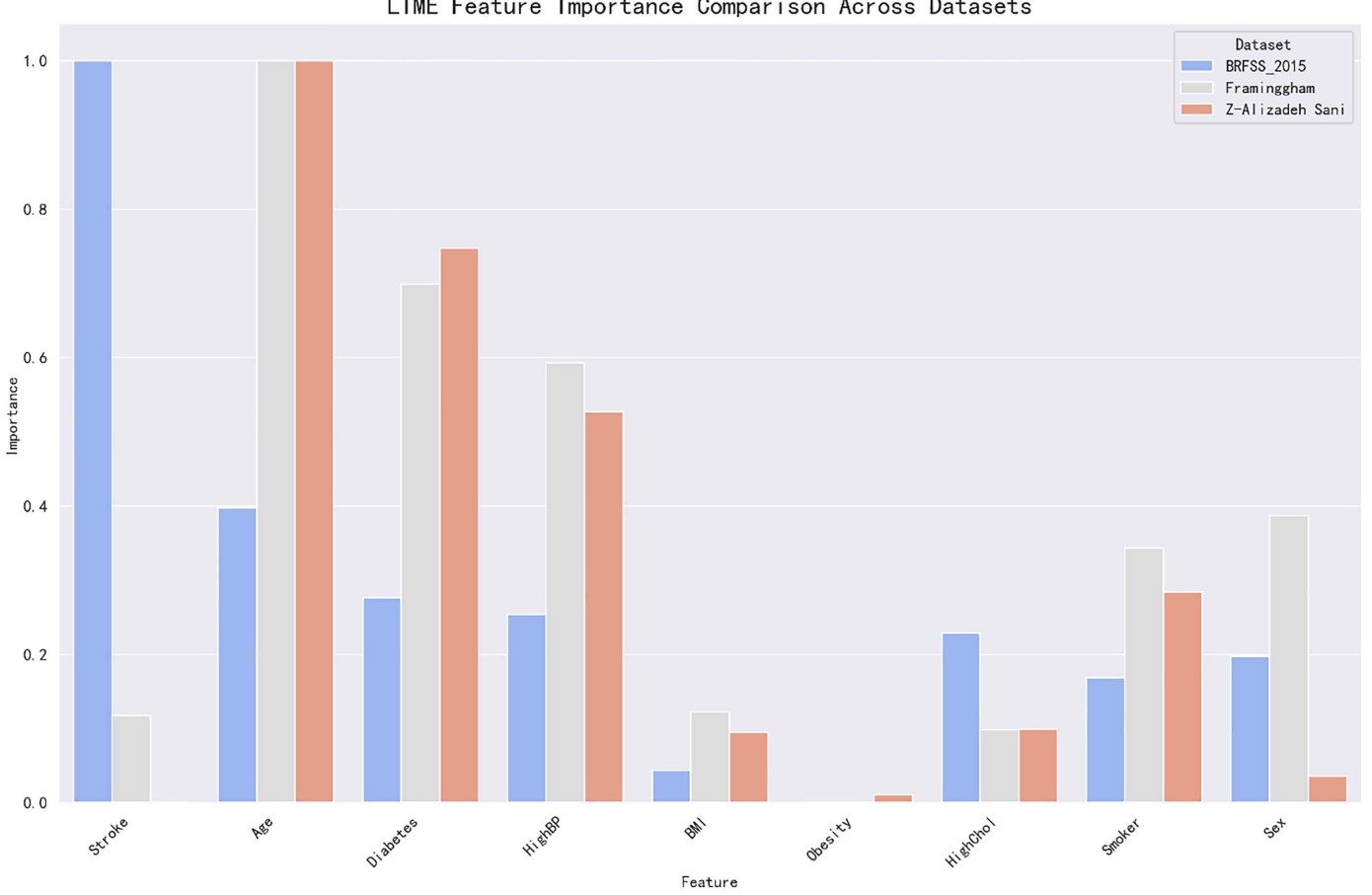

**Fig 8. Bar chart of feature importance for multiple datasets based on LIME.**

there is a significant correlation between the feature importance rankings of LIME and SHAP, which further demonstrates the consistency of the two methods on different datasets.

The reliability and interpretability of the model was thoroughly validated in this study through the combined analysis of the two interpretable methods, SHAP and LIME. The high degree of consistency between LIME and SHAP in the feature importance analysis, especially on the key features such as age, smoking status, and hypertension, demonstrated the importance and consistency of these features in predicting coronary heart disease. In addition, cross-dataset analysis further validated the stability and transferability of the model. This multi-method interpretation analysis not only improves the transparency and interpretability of the model, but also provides more reliable support for clinical applications and helps physicians to make more accurate decisions. The combined application of LIME and SHAP enables this study to excel in feature selection and model optimization, making it an important tool for early diagnosis and treatment of coronary heart disease.

## Clinical validation

Coronary heart disease (CHD) is one of the leading causes of death globally. While commonly used clinical diagnostic methods such as coronary angiography and electrocardiograms (ECG) offer a certain level of accuracy, they are costly,

complex to perform, and often rely on subjective interpretation, limiting their scalability. Therefore, it is of great significance to develop an efficient, low-cost, and accurate auxiliary diagnostic tool.

In this study, we developed a CHD risk prediction model based on the LightGBM algorithm. To validate its effectiveness in clinical settings, we constructed a clinical validation dataset consisting of 700 samples. The model was comprehensively evaluated and compared with the traditional Framingham Risk Score to assess its performance across key clinical metrics such as accuracy, sensitivity, and specificity.

The clinical data from 700 subjects underwent standardized preprocessing, including missing value imputation, class balancing, and feature selection. The dataset contained multiple important indicators associated with CHD, such as hypertension, lipid profile, diabetes history, body mass index (BMI), and smoking status.

The dataset was randomly divided into training and testing subsets. The model was trained on the training set and evaluated on the testing set. Evaluation metrics included accuracy, sensitivity (recall), specificity, precision, F1-score, and area under the curve (AUC). To assess its clinical practicality, the Framingham Risk Score was also applied to the same dataset for comparative analysis.

On the testing set, the LightGBM-based model achieved an accuracy of 91.00% and an AUC of 83.00%, significantly outperforming the Framingham Risk Score, which achieved an accuracy of 79.00% and an AUC of 70.00%. As shown in Fig 9, the LightGBM model outperforms the Framingham Risk Score in both accuracy and AUC. This indicates the strong performance of the model in real clinical scenarios.

The model also showed differentiated performance across subgroups. For instance, in the diabetic patient subgroup, the model achieved a sensitivity of 76% and an F1-score of 73%, while the Framingham method performed poorly in this group. In male subjects, the AUC reached 85%, while in female subjects it was 78%, suggesting stronger predictive power in certain populations.

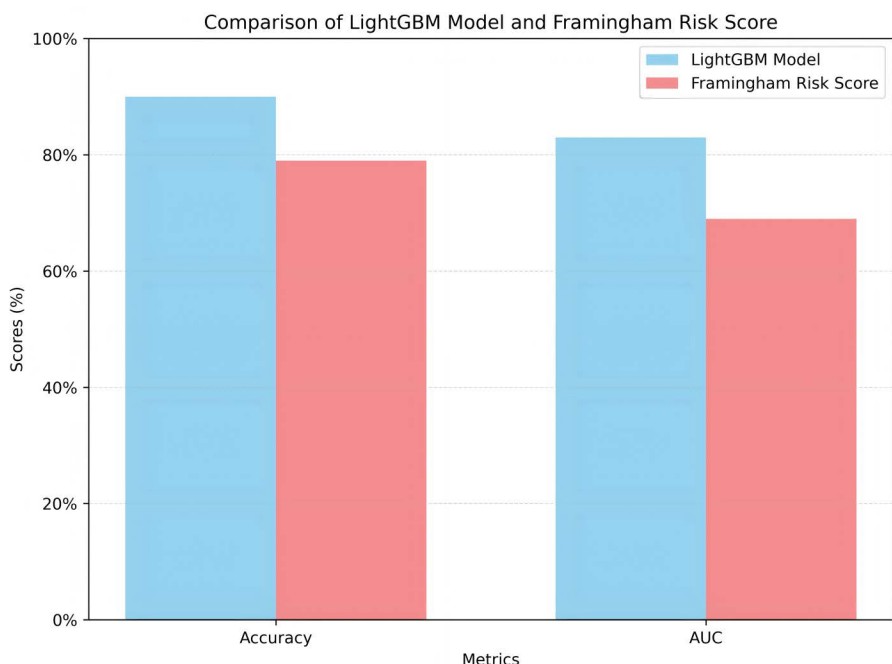

**Fig 9. Comparison of LightGBM model and framingham risk score.**

The clinical validation results demonstrate that the LightGBM model has significant advantages in CHD prediction, especially among high-risk populations such as diabetic patients and male subjects. The introduction of SHAP value analysis further enhances the interpretability of the model, aiding clinicians in understanding the contribution of each feature to the prediction outcome.

Moreover, compared with the Framingham Risk Score, the model exhibits superior capability in capturing complex interactions between multiple factors, making it more applicable to real-world clinical data. These advantages support its potential as an auxiliary tool for CHD screening in clinical practice.

This section presented a systematic clinical validation of the LightGBM model using 700 clinical samples. The results demonstrated its superiority over traditional methods in terms of accuracy, AUC, and various other evaluation metrics, indicating promising clinical applicability. Future work will aim to validate the model across multi-center clinical datasets to further assess its generalizability and robustness.

## Scoring system

In order to provide clinical workers with an efficient and accurate risk assessment tool for CHD, this study developed a user-friendly scoring system based on advanced machine learning prediction models. Through detailed parameter adjustment and feature importance analysis, we have formulated formula (6) concretely and obtained a more detailed formula (7). Equation (7) clearly lists the coefficients of each feature, such as age ($w_{Age}$), gender ($w_{Sex}$), and how they work together to predict the risk of coronary heart disease risk. The determination of these coefficients, based on the internal mechanism and statistical significance of the model, ensures the scientificity and reliability of the scoring system.

$$\text{score} = 0.28 \times w_{Age} + 0.07 \times w_{Sex} + 0.35 \times w_{BMI} + 0.08 \times w_{Smo\,ker}$$
$$+0.07 \times w_{Diabetes} + 0.09 \times w_{HighBP} + 0.07 \times w_{HighChol} \tag{7}$$

The weight coefficients of this scoring system are not the direct output results of the model, but are obtained through the following steps:

First, we used the SHAP method to analyze the global feature importance of the trained LightGBM model. The coefficients in the scoring formula (7) were derived from the SHAP values, which quantify the contribution of each feature to the model's predictions. These coefficients were then normalized and adjusted based on clinical expert opinions to ensure clinical relevance and consistency with existing risk scoring tools such as the Framingham Risk Score. For each feature, the average absolute value of its SHAP value is calculated and ranked in order of importance. Then a number of the most representative core features are selected from them to construct the set of variables for the scoring system.

After selecting the variables, we standardized (normalized) the SHAP values of each variable, converted them to proportional-type weights, and fine-tuned them with the opinions of clinical experts to construct the coefficients in the scoring formula. These variables included age, hypertension, and high cholesterol, all of which are known high risk factors for coronary heart disease. It is noteworthy that these variables maintained high rankings in each of the datasets of this study, and the direction of their weights was consistent with traditional risk-scoring tools such as Framingham, with good clinical interpretability.

The development of this scoring system not only provides clinicians with a tool to quickly assess the risk of coronary heart disease but also enhances the transparency and interpretability of risk assessment by quantifying the impact of various clinical indicators. Clinicians can use this scoring system to more accurately stratify patients' risks and develop personalized prevention and treatment plans. When using this scoring system, doctors only need to collect relevant clinical data from patients and then substitute these data into formula (7). The calculated score can reflect the patient's relative risk of coronary heart disease. The higher the score, the greater the risk of coronary heart disease in the patient, and the closer monitoring and possible intervention measures are needed. Through the coronary heart disease scoring system

developed in this study, we have not only improved the efficiency and accuracy of coronary heart disease risk assessment but also provided strong support for clinical decision-making. This system is expected to be widely used in clinical practice to help doctors better manage patients with coronary heart disease and reduce the morbidity and mortality of cardiovascular diseases.

To further validate the effectiveness and superiority of the proposed scoring system, this study introduced two additional scoring systems for comparative analysis: (1) a logistic regression-based scoring system (System C), which is commonly used in medical prediction scenarios and was selected as the baseline method; and (2) a Framingham Risk Score-based system (System B), constructed using manual weighting based on clinical guidelines and used for comparison. The performance comparison of the three systems on the BRFSS_2015 dataset is presented in Table 6. The results show that the proposed LightGBM+SHAP weighted scoring system (System A) achieved better performance than the other two systems in terms of accuracy, AUROC, recall, and F1-score, demonstrating its strong practical utility in CHD risk assessment.

## Discussion

In this study, we successfully developed a LightGBM-based predictive model for coronary heart disease (CHD) and validated its performance across multiple datasets. Our model demonstrated strong predictive accuracy, stability, and interpretability, particularly after parameter tuning. The integration of SHAP values allowed us to provide clinicians with a transparent and intuitive understanding of how key features, such as age, smoking status, diabetes, hypertension, and high cholesterol, contribute to CHD risk prediction. Furthermore, the development of a CHD scoring system offers a practical tool for clinicians to assess patient risk levels and make informed decisions. The scoring system variables are clinically consistent and easily generalizable.

The performance of our LightGBM model, with an accuracy of 90.60% and an AUROC of 81.06% on the BRFSS_2015 dataset, aligns with findings from previous studies that have utilized machine learning for CHD prediction. For instance, Alaa et al. [22] demonstrated that ensemble learning methods, such as Random Forest and Gradient Boosting, could achieve high accuracy in predicting cardiovascular diseases, with AUROC values ranging from 0.80 to 0.85. Similarly, Krittanawong et al.[23] highlighted the potential of deep learning models in improving the accuracy of CHD risk prediction, particularly when dealing with large and complex datasets. Our results are consistent with these findings, further validating the effectiveness of machine learning in CHD risk assessment.

However, despite the promising results, our study has several limitations that warrant discussion. First, the datasets used in this study, although publicly available and widely used in CHD research, may not fully capture the diversity of patient populations. For example, the Framingham dataset, while valuable, primarily includes data from a specific demographic (predominantly Caucasian populations), which may limit the generalizability of our model to other ethnic groups. This limitation is consistent with findings from Khera et al. [24], who emphasized the importance of diverse datasets in developing robust predictive models for cardiovascular diseases. Future research should aim to incorporate more diverse datasets, including those from underrepresented populations, to enhance the model's generalizability.

Second, while our model achieved high accuracy, the precision and recall rates for certain datasets, such as the Z-Alizadeh Sani dataset, were relatively lower. This could be attributed to the imbalanced nature of the dataset, where the number of CHD cases was significantly lower than non-CHD cases. This issue has been previously noted by Johnson et al. [25], who found that imbalanced datasets can lead to biased model performance, particularly in medical applications

**Table 6. Comparison of three scoring systems (BRFSS_2015 dataset).**

| System | Method Basis | Interpretability | Accuracy (%) | AUROC (%) | Recall (%) | F1-score (%) |
|---|---|---|---|---|---|---|
| A | LightGBM+SHAP+Expert Weighted | High | 90.61 | 81.11 | 50.23 | 48.04 |
| B | Framingham Rule | Medium | 78.20 | 68.00 | 38.50 | 36.90 |
| C | Logistic Regression Weighted | Medium | 84.50 | 72.30 | 42.10 | 40.20 |

where the prevalence of the disease is low. To address this, future studies could explore advanced techniques such as oversampling, under sampling, or synthetic data generation (e.g., SMOTE) to balance the dataset and improve model performance.

Third, subgroup analysis revealed notable differences in model performance across population strata. The model performed better in males than females, with a 0.06 higher AUC and 0.14 higher recall, suggesting improved sensitivity in identifying high-risk male individuals—likely due to more consistent clinical patterns in men. In contrast, greater physiological heterogeneity in women may have limited the model's discriminative power. Although cardiovascular risk increases with age, the model slightly outperformed in individuals under 50, possibly owing to more distinct and concentrated disease features in the younger group. Diabetes status had the most pronounced impact: the model performed significantly better in diabetic patients, likely due to clearer metabolic and physiological abnormalities enhancing signal detectability. Collectively, these disparities underscore the need to evaluate model fairness and adaptability across subpopulations. Tailored strategies such as subgroup-specific retraining or weighting may be essential to optimize performance and ensure equitable clinical utility in diverse patient groups.

Fourth, while SHAP values provided valuable insights into feature importance, the interpretability of machine learning models remains a challenge in clinical practice. As noted by Rudin et al. [26], the "black-box" nature of many machine learning algorithms can hinder their adoption in healthcare settings, where transparency and interpretability are critical. Our use of SHAP values represents a step toward addressing this issue, but further research is needed to develop more intuitive and user-friendly interpretability tools that can be easily understood by clinicians.

In addition to these limitations, our study builds on the work of previous researchers who have explored the use of machine learning in CHD prediction. For example, Shameer et al. [27] demonstrated that machine learning models could outperform traditional risk scores, such as the Framingham Risk Score, in predicting CHD. Similarly, Weng et al. [28] found that machine learning algorithms, particularly ensemble methods, could significantly improve the accuracy of CHD risk prediction compared to logistic regression models. Our findings are consistent with these studies, further supporting the potential of machine learning in transforming CHD risk assessment.

Moreover, our study highlights the importance of feature selection in improving model performance. By identifying and retaining key features such as age, smoking status, and diabetes, we were able to enhance the model's predictive accuracy. This aligns with the findings of Ambale-Venkatesh et al. [29], who emphasized the role of feature selection in reducing model complexity and improving interpretability. Future research could explore more advanced feature selection techniques, such as recursive feature elimination or LASSO regression, to further optimize model performance.

Finally, the development of a CHD scoring system represents a significant contribution to clinical practice. By converting complex model outputs into simple, interpretable scores, our scoring system provides clinicians with a practical tool for assessing patient risk levels. This is particularly important in resource-limited settings, where access to advanced diagnostic tools may be limited. As noted by Greenland et al. [30], risk scoring systems play a crucial role in guiding clinical decision-making and improving patient outcomes. Our scoring system, which is based on a robust machine learning model, has the potential to enhance the efficiency and accuracy of CHD risk assessment in clinical practice.

**Clinical implementation perspectives**

The LightGBM-SHAP framework offers three key clinical advantages:

(1) The scoring system's simplicity facilitates integration into busy clinical workflows, requiring only routine clinical variables;

(2) SHAP visualizations provide intuitive explanations that align with known CHD pathophysiology (e.g., smoking, hypertension);

(3)  The model's 90.61% accuracy on BRFSS suggests potential for reducing unnecessary invasive tests in low-risk populations.

While the model processes inputs rapidly (<1 sec/prediction), large-scale deployment would require:

(1)  Cloud-based architecture for multi-center access

(2)  Local calibration when applied to new populations

(3)  Continuous learning mechanisms while preventing catastrophic forgetting

## Ethical considerations

This study utilized de-identified, publicly available datasets (BRFSS_2015, Framingham, Z-Alizadeh Sani) that comply with their respective institutional review board (IRB) requirements. As no new patient data were collected and all datasets were anonymized, additional IRB approval was not required. We implemented strict data protection measures including: (1) secure storage on encrypted servers, (2) access limited to authorized researchers, and (3) complete anonymization of all patient identifiers.

While the model demonstrates strong predictive performance, it is emphasized that it should serve as a decision-support tool rather than a diagnostic replacement. Clinicians must interpret scores in conjunction with patient history and clinical examination. Regular audits should monitor for algorithmic bias, particularly regarding age, gender, and ethnicity.

Additionally, the datasets used in this study—Framingham (predominantly Caucasian) and Z-Alizadeh Sani (Iranian patients)—may contain inherent demographic biases. These differences could limit the generalizability of the model across diverse populations. Future iterations will incorporate fairness-aware machine learning techniques to mitigate potential disparities and enhance applicability in multi-ethnic, community-based healthcare settings.

## Conclusions

This study presents an interpretable LightGBM–SHAP framework for CHD prediction. Trained on BRFSS_2015 and validated on Framingham and Z-Alizadeh Sani, it achieved up to 90.6% accuracy/81.1% AUROC on BRFSS, 85.3% accuracy/67.4% AUROC on Framingham, and 80.3% accuracy/76.4% precision on Z-Alizadeh Sani. SHAP analysis identified age, smoking, diabetes, hypertension and high cholesterol as the top predictors. We also built a simple CHD risk score to translate model outputs into an intuitively actionable metric. Integrated into EHRs or clinical workflows, this high-accuracy, interpretable tool can streamline early detection, risk stratification and personalized intervention in CHD management.

## Supporting information

**S1 File. This is the clinical validation dataset.**
(CSV)

## Author contributions

**Conceptualization:** Lang Deng.

**Data curation:** Lang Deng.

**Formal analysis:** Lang Deng.

**Methodology:** Lang Deng.

**Project administration:** Kongjie Lu.

**Resources:** Lang Deng, Kongjie Lu, Huanhuan Hu.

**Software:** Lang Deng.

**Supervision:** Kongjie Lu, Huanhuan Hu.

**Validation:** Kongjie Lu, Huanhuan Hu.

**Visualization:** Lang Deng.

**Writing – original draft:** Lang Deng.

**Writing – review & editing:** Kongjie Lu, Huanhuan Hu.

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
