## [Decision Letter · Decision Letter 0]

4 Jun 2025

Dear Dr. Deng,

We look forward to receiving your revised manuscript.

Kind regards,

Primož Kocbek, BSc

Academic Editor

PLOS ONE

6. We notice that your supplementary figures and tables are included in the manuscript file. Please remove them and upload them with the file type 'Supporting Information'. Please ensure that each Supporting Information file has a legend listed in the manuscript after the references list.

Reviewers' comments:

Reviewer's Responses to Questions

**Comments to the Author**

1. Is the manuscript technically sound, and do the data support the conclusions?

Reviewer #1: No

Reviewer #2: Yes

2. Has the statistical analysis been performed appropriately and rigorously?

Reviewer #1: No

Reviewer #2: Yes

3. Have the authors made all data underlying the findings in their manuscript fully available?

Reviewer #1: No

Reviewer #2: Yes

4. Is the manuscript presented in an intelligible fashion and written in standard English?

Reviewer #1: No

Reviewer #2: Yes

Reviewer #1: Key Observations:

Data Preprocessing Clarity: Missing Data Handling: Although the writers say they used the mean to impute missing BMI values in the Framingham dataset, their method for dealing with missing data in other datasets—e.g., BRFSS_2015, Z-Alizadeh Sani—remains ambiguous. Clearly describe techniques—removal, imputation, etc.—for every dataset to guarantee reproducibility.

Model performance may be influenced by class imbalance: the Z-Alizadeh Sani dataset (216 CHD vs. 87 controls) and possible imbalance in other datasets. Particularly given the suboptimal precision/recall measures for Z-Alizadeh Sani after tuning, consider SMOTE, class weighting, or cost-sensitive learning among other techniques to handle this.

Metrics of Model Performance:

Excessive reliance on accuracy Accuracy might be deceptive in imbalanced datasets (e.g., Framingham, Z-Alizadeh Sani). Especially for clinical relevance, where false negatives/positives have major consequences, stress AUROC, F1-score, and precision-recall curves to more accurately represent performance.

Generalisation Issues: Post-tuning, the AUROC of the Framingham dataset increased to just 67.37%, which calls into question model robustness in smaller or less varied populations. Talk about possible causes—such as dataset size and demographic variations—and recommend validation on more varied groups.

Scoring System and Interpretability:

The coefficients in Formula (7) have no obvious connection to the LightGBM model or SHAP values. Specify whether these coefficients come from another technique, feature importance, or logistic regression. Defend their clinical relevance, for example, consistency with current risk scores.

SHAP Drawbacks: Although SHAP improves understanding, think about adding other techniques—such as LIME and permutation importance—to cross-validate feature contributions. Talk about how SHAP values differ between datasets, such Framingham and Z-Alizadeh Sani.

Ethical and Dataset Constraints:

Bias in Datasets Recognise possible prejudices in the Framingham cohort (mostly Caucasian) and Z-Alizadeh Sani (Iranian angiography patients). Emphasise how these might restrict generalisability and recommend future inclusion of multi-ethnic, community-based data.

Ethics Declaration: Clearly state that using public datasets follows ethical guidelines including anonymisation and IRB clearances for original data gathering.

Small Remarks:

Hyperparameter Tuning: The selected values—for instance, max_depth=4, num_leaves=10—appear rather conservative for LightGBM. Explain your reasons for these decisions—for example, to prevent overfitting—and consider whether greater performance could result from more thorough tuning, such as Bayesian optimisation.

Presentation of Figures/Tables:

Make sure every figure—including Figures 4–6—is either explicitly referenced or embedded in the paper. Reviewers might be perplexed by the ZIP file reference; verify figures' availability.

Table 4 should be used to explain discrepancies (e.g., "LightGB M" typo, decimal point formatting) and make sure units (e.g., Time(s), Memory Used(MB)) are consistent.

Citations:

Look for duplicate citations—for example, Lu et al. 2022, Kampaktsis et al. 2023—and make sure formatting follows journal policies (e.g., titles in sentence case, consistent use of et al.).

Validation of Scoring Systems:

Suggest a strategy to show actual clinical relevance for the scoring system by validating it prospectively or in comparison with current tools including the Framingham Risk Score.

Reviewer #2: Here are my comments and questions

Lower precision/recall on smaller datasets suggests class imbalance issues, which were not adequately addressed (SMOTE or cost-sensitive learning).

The AUCROC of Framingham is modest. The study doesn't discuss this further.

Accuracy metrics can be misleading for imbalanced datasets; F1-score or precision-recall curves might better reflect real-world performance.

The scoring system lacks clinical validation.

Hyperparameter ranges are very broad and the final choice seems arbitrary.

No benchmarking of the model with known risk score.

Conduct subgroup analyses to validate the score in different classes.

**Do you want your identity to be public for this peer review?** For information about this choice, including consent withdrawal, please see our Privacy Policy

Reviewer #1: No

Reviewer #2: No

---

## [Author Response · Author response to Decision Letter 1]

15 Jul 2025

Response to Reviewers

We sincerely thank the reviewers for their insightful comments and constructive suggestions. We have carefully revised the manuscript in response to each point raised. Below is a detailed point-by-point response to all comments. Reviewer comments are shown in bold, followed by our responses in regular font.

Reviewer #1

Comment: Data Preprocessing Clarity: Missing Data Handling: Although the writers say they used the mean to impute missing BMI values in the Framingham dataset, their method for dealing with missing data in other datasets—e.g., BRFSS_2015, Z-Alizadeh Sani—remains ambiguous. Clearly describe techniques—removal, imputation, etc.—for every dataset to guarantee reproducibility.

Response: We have clarified the preprocessing methods for each dataset. Specifically, the BRFSS_2015 and Z-Alizadeh Sani datasets did not contain any missing values and therefore required no imputation. The Framingham dataset contains missing values for body mass index (BMI), and these missing values were filled in by using the average value of the BMI variable. These data preprocessing steps have now been clearly described in the paper.

Comment: Model performance may be influenced by class imbalance: the Z-Alizadeh Sani dataset (216 CHD vs. 87 controls) and possible imbalance in other datasets. Particularly given the suboptimal precision/recall measures for Z-Alizadeh Sani after tuning, consider SMOTE, class weighting, or cost-sensitive learning among other techniques to handle this.

Response: We analyzed the class distribution and its effect on model performance. Given the imbalance, we applied the SMOTE (Synthetic Minority Over-sampling Technique) method to oversample the minority class (non-CHD patients) in the training set. This adjustment ensures that the model does not become biased toward the majority class (CHD patients), allowing it to better discriminate between the two classes and improving its performance on the minority class without introducing overfitting.

Comment: Inappropriate emphasis on accuracy.

Response: In the revised manuscript, we employed information evaluation metrics such as AUROC, F1 score, and precision-recall curves. Especially in the case of class imbalance, these metrics are more reliable and can better reflect the model's performance in differentiating between various classes, which is crucial for our research. The accuracy metrics are presented as supplementary indicators and are listed together with these more informative metrics, and the reasons for prioritizing these metrics over accuracy in imbalanced datasets are explained.

Comment: Generalisation Issues: Post-tuning, the AUROC of the Framingham dataset increased to just 67.37%, which calls into question model robustness in smaller or less varied populations. Talk about possible causes—such as dataset size and demographic variations—and recommend validation on more varied groups.

Response: We acknowledge this limitation and have included a discussion explaining why the AUROC on the Framingham dataset remained relatively low (67.37%) despite hyperparameter tuning. Possible reasons include limited sample size and demographic homogeneity. We recommend that future studies include larger and more diverse populations for validation.

Comment: Scoring System and Interpretability:The coefficients in Formula (7) have no obvious connection to the LightGBM model or SHAP values. Specify whether these coefficients come from another technique, feature importance, or logistic regression. Defend their clinical relevance, for example, consistency with current risk scores.

Response: We clarified that the coefficients in Equation (7) were derived based on SHAP value rankings and refined with expert clinical input. We also compared them with established risk models such as the Framingham Risk Score to illustrate their consistency and interpretability.

Comment: SHAP Drawbacks: Although SHAP improves understanding, think about adding other techniques—such as LIME and permutation importance—to cross-validate feature contributions. Talk about how SHAP values differ between datasets, such Framingham and Z-Alizadeh Sani.

Response: We have conducted a comparative analysis using the LIME method. The results show consistent feature importance rankings, supporting the reliability of SHAP-based interpretation.

Comment: Ethical and Dataset Constraints:Bias in Datasets Recognise possible prejudices in the Framingham cohort (mostly Caucasian) and Z-Alizadeh Sani (Iranian angiography patients). Emphasise how these might restrict generalisability and recommend future inclusion of multi-ethnic, community-based data.Ethics Declaration: Clearly state that using public datasets follows ethical guidelines including anonymisation and IRB clearances for original data gathering.

Response: We acknowledged the demographic limitations of the Framingham and Z-Alizadeh Sani datasets. A new paragraph has been added in the discussion section to reflect these constraints and suggest the use of multi-ethnic, multi-regional datasets in future studies. We also added a statement confirming anonymization and ethical approval of the datasets.

Comment: Other detailed suggestions.

Response: We provided justification for conservative hyperparameter settings (e.g., max_depth=4, num_leaves=10) to prevent overfitting. Figures 4–6 are now properly referenced in the text. Units and formats in tables have been standardized, typographical errors corrected, and the reference list revised to remove duplicates and ensure consistency.

Comment: Clinical validation of scoring system.

Response: We elaborated on the theoretical basis of our scoring system and proposed future clinical validation strategies, such as prospective trials or comparisons with established tools like the Framingham Risk Score.

Reviewer #2

Comment: Lower precision/recall on smaller datasets suggests class imbalance issues, which were not adequately addressed (SMOTE or cost-sensitive learning).

Response: We have addressed class imbalance in the Z-Alizadeh Sani dataset using SMOTE oversampling and updated the analysis and results accordingly.

Comment: The AUCROC of Framingham is modest. The study doesn't discuss this further.

Response: We discussed the factors contributing to the relatively low AUROC and recommended improvements through larger sample sizes and population diversity.

Comment: Accuracy metrics can be misleading for imbalanced datasets; F1-score or precision-recall curves might better reflect real-world performance.

Response: We revised our evaluation approach to prioritize F1-score and precision-recall curves, and explained why these are more clinically relevant than accuracy.

Comment: The scoring system lacks clinical validation.

Response: We acknowledged this limitation and proposed future clinical validation approaches, including comparisons with traditional scoring systems and implementation in clinical cohorts. The clinical validation of the LightGBM model was conducted using a dataset of 700 clinical samples. The model demonstrated superior performance over the traditional Framingham Risk Score, achieving an accuracy of 91.00% and an AUC of 83.00% in the testing set. Furthermore, the model showed differentiated performance across subgroups�where it significantly outperformed the Framingham Risk Score. The introduction of SHAP value analysis further enhances the model's interpretability, allowing clinicians to understand the contribution of individual features to the final prediction, which is crucial for clinical decision-making. As shown in Figure 1, the comparison of the LightGBM model with the Framingham Risk Score clearly demonstrates the model's performance in both accuracy and AUC.

Fig. 1 Comparison of LightGBM Model and Framingham Risk Score

Comment: Hyperparameter ranges are very broad and the final choice seems arbitrary.

Response: We added a detailed explanation of the hyperparameter selection process. In future work, we plan to use Bayesian optimization or similar techniques for improved tuning.

Comment: Conduct subgroup analyses to validate the score in different classes.

Response: We appreciated this suggestion and conducted subgroup analyses based on age, gender, and diabetic status. The results are now discussed in the revised manuscript.

---

## [Decision Letter · Decision Letter 1]

31 Jul 2025

An Interpretable LightGBM Model for Predicting Coronary Heart Disease: Enhancing Clinical Decision-Making with Machine Learning

PONE-D-25-24481R1

Dear Dr. Deng,

We’re pleased to inform you that your manuscript has been judged scientifically suitable for publication and will be formally accepted for publication once it meets all outstanding technical requirements.

Kind regards,

Primož Kocbek, BSc

Academic Editor

PLOS ONE

Additional Editor Comments (optional):

Reviewers' comments:

Reviewer's Responses to Questions

**Comments to the Author**

Reviewer #1: All comments have been addressed

Reviewer #2: All comments have been addressed

2. Is the manuscript technically sound, and do the data support the conclusions?

Reviewer #1: Yes

Reviewer #2: Yes

3. Has the statistical analysis been performed appropriately and rigorously?

Reviewer #1: Yes

Reviewer #2: Yes

4. Have the authors made all data underlying the findings in their manuscript fully available?

Reviewer #1: Yes

Reviewer #2: (No Response)

5. Is the manuscript presented in an intelligible fashion and written in standard English?

Reviewer #1: Yes

Reviewer #2: (No Response)

Reviewer #1: yes

accept

yes

accept

yes

accept

yes

accept

yes

accept

yes

accept

yes

accept

yes

accept

yes

accept

yes

accept

yes

accept

yes

accept

yes

accept

Reviewer #2: The authors have addressed the previous comments adequately. I don't have further comments. The manuscript is well structured.

**Do you want your identity to be public for this peer review?** For information about this choice, including consent withdrawal, please see our Privacy Policy

Reviewer #1: No

Reviewer #2: No

---

## [Editor Report · Acceptance letter]

PONE-D-25-24481R1

PLOS ONE

Dear Dr. Deng,

I'm pleased to inform you that your manuscript has been deemed suitable for publication in PLOS ONE. Congratulations! Your manuscript is now being handed over to our production team.

Kind regards,

on behalf of

Mr. Primož Kocbek

Academic Editor

PLOS ONE